# Changes in Jumping and Throwing Performances in Age-Group Athletes Competing in the European Masters Athletics Championships between 1978 and 2017

**DOI:** 10.3390/ijerph16071200

**Published:** 2019-04-03

**Authors:** Alexandra M. L. Kundert, Pantelis T. Nikolaidis, Stefania Di Gangi, Thomas Rosemann, Beat Knechtle

**Affiliations:** 1Institute of Primary Care, University of Zurich, 8006 Zurich, Switzerland; a.kundert@kundert.ch (A.M.L.K.); stefania.digangi@usz.ch (S.D.G.); thomas.rosemann@usz.ch (T.R.); 2Exercise Physiology Laboratory, 18450 Nikaia, Greece; pademil@hotmail.com; 3Medbase St. Gallen Am Vadianplatz, 9001 St. Gallen, Switzerland

**Keywords:** age, athletics, master athlete, age group, sex, track and field

## Abstract

The results of master athletes have been used previously to examine the age-related differences in aerobic capacity, however, less research has been conducted on the variation of jumping and throwing performances with aging. Therefore, the aim of the present study of elite master athletes was to investigate (a) the age-related differences in throwing (i.e., discus, hammer, javelin, and shot put) and jumping events (i.e., high jump, long jump, pole vault, and triple jump) in 5-year age-group intervals from 35–39 to 95–99 years of elite master athletes, and (b) the trends in performance and sex differences. The top eight female and male finalists for each age group and in each event from 20 European Masters Athletics Championships held between 1978 and 2017 were considered. Overall, 13,673 observations from 4726 master athletes were analyzed. For each event separately, a mixed regression model was performed with sex, age group, calendar year, and interaction terms (sex-age group, sex-year) defined as fixed effects. Performances were improving over time with a linear trend overall for each event. Men had better performances as compared to women, (i.e., in triple jump the estimated difference was 2.58 m, *p* < 0.001). Performances declined with age for each event (i.e., in triple jump, compared with the age group 45–49 years, performance in the age group 35–39 years was 0.98 m better and performance in the age group 85–89 years was 6.24 m worse). The decline of male performances with age was either slower or faster than the decline of female performances depending on age groups and events.

## 1. Introduction

Aging in humans has been associated with a decline in a series of physical fitness components such as muscle power. Accordingly, reduced muscle power has resulted in poor performance in the daily tasks of the elderly [1,2]. To manage this poor performance, it is necessary to understand the age-related differences in physical fitness and the beneficial role of exercise. Master athletes of sports relying on muscle power present an optimal model to use for the study of this topic.

Recent studies of running events have shown that master marathoners exhibit a relatively smaller decrease in aerobic capacity with aging as compared to nonathletes [3]. In addition, the effect of aging on physical fitness components may vary by sex. For instance, a study investigating the performance in various running race distances (i.e., 100 m, 200 m, 400 m, 800 m, 1500 m, 5000 m, 10,000 m, and marathon) in the European Masters Athletics Championships held between 1978 and 2014 showed that both male and female athletes improved their performances over time, and that the sex gap may have reached its limit with the differences between the sexes being greater in the shorter running distances [4]. Nevertheless, limited information exists with regards to the age-related differences in jumping and throwing performances in master athletes. In a recent study investigating jumping and throwing performances in master athletes competing in the World Masters Athletics (WMA) Championships from 1975 to 2016, men performed better than women, but women improved their performance more than men over time. Furthermore, the decline in performance with age was dependent on both sex and event [5].

It is well known that male athletes perform better than female athletes [6] in different sports disciplines such as swimming [7] and running [8]. Little is known, however, about the sex differences in performance with increasing age and the differences between different sports disciplines (i.e., swimming versus running) [9]. Recent studies investigated master swimmers competing at a world class level in the FINA World Championships (Fédération Internationale de Natation) and showed that women in older age groups reduced the gap between women and men and that they were able to achieve a similar performance to men in older age groups for different strokes such as freestyle [10], backstroke [11], breaststroke [12], butterfly [13], individual medley [14], and open-water, long-distance swimming [15].

In addition to endurance events such as running and swimming, jumping and throwing events also attract the interest of master athletes. Master athletes of throwing and jumping events present an appropriate model for the study of the effect of aging on muscle power. It is well known that jumping performance decreases with increasing age in athletes [16]. So far, the research in jumping and throwing events has focused on master world records [17,18]. A linear decline in performance over 70 years has been highlighted with a 1.1–1.4% annual decrease in performance [17]. Furthermore, an inter-event variation of this decline has been shown with the throwing and pole vault events presenting the greatest rates of decline with aging [18].

An existing gap in the research literature is the lack of research in master athletes of jumping and throwing events using larger samples than in the past [17,18]. For example, Gava et al. [17] in 2015, investigated the World records in running events by the end of 2012 in 100 m, 200 m, 400 m, 800 m, 1500 m, 3000 m, 5000 m, and 10,000 m for athletes from 35 to 100 years of age. Baker et al. [18] analyzed the masters athletic records in track and field events, published in September 1999, for mature athletes over the age of 30 years by decade. The track events were divided into the sprint events (i.e., 100 m, 200 m, and 400 m), middle distance events (i.e., 800 m, 1500 m, and 3000 m), long distance events (i.e., 5000 m, 10,000 m, and marathon), hurdling events (i.e., 110 m, 400 m, and steeple chase) and walking events. The jumping events (i.e., long jump, high jump, triple jump, and pole vault) were separated from the throwing events (i.e., shot put, discus, javelin, and hammer).

Moreover, the effects of sex differences and how they vary by sport event (e.g., jumping events), calendar year, and age group have not yet been examined. The knowledge of this information would be important from both a theoretical and a practical perspective. Coaches and fitness trainers working with master athletes in jumping and throwing events could set age-tailored training goals and develop training programs, accordingly. On the other hand, geriatrists and sports scientists focusing on the relationship of aging with physical fitness and physical activity could enhance their understanding of the interplay among these parameters.

Therefore, the aim of the present study was to investigate differences in performance by age and calendar year for master athletes competing in jumping (i.e., high jump, long jump, triple jump, and pole vault) and throwing (i.e., hammer throw, javelin throw, discus throw, and shot put) events in 20 European Masters Athletics Championships held since 1978. Sex differences in performance were examined for jumping events. On the basis of recent findings for runners competing in different distances in the European Masters Athletics Championships [4], an improvement in performance over years and a variation of sex differences among events were expected. In addition, the muscle performance of upper and lower limbs (higher in the latter one) and the force-velocity characteristics (upper limbs rely relatively more on velocity than on force) differed [19,20]. Since force declines more than velocity with aging [21,22], it was hypothesized that throwing events would exhibit larger age-related differences than jumping events.

## 2. Materials and Methods

### 2.1. Experimental Approach to the Problem

To investigate the age-related differences in throwing events (i.e., discus, hammer, javelin, and shot put) and jumping events (i.e., high jump, long jump, pole vault, and triple jump), as well as the performance trends and the sex differences in performance of elite master athletes, we analyzed the competition results of the European Masters Athletics Championships for master athletes competing in 5-year age-group intervals from 35–39 to 95–99 years of age. All data were gathered from the website of the European Masters Athletics, www.european-masters-athletics.org. The European Masters Athletics organizational goals are to manage and control athletics for masters, to conduct the European Masters Athletics Championships, and to record the results achieved. Master athletes are defined as women and men who are at least 35 years of age. This study was approved by the Institutional Review Board of Kanton St. Gallen, Switzerland, with a waiver of the requirement for informed consent of the participants as the study involved the analysis of publicly available data (01 June 2010).

### 2.2. Subjects

In the present study, data from 20 European Masters Athletics Championships, which took place between 1978 and 2017, were included. Events such as high jump, long jump, triple jump, pole vault, hammer throw, javelin throw, discus throw, and shot put were explored. Only data from outdoor championships were considered, since the different conditions for indoor championships might lead to slightly different results. The master athletes were categorized into 5-year age-group intervals from 35–39 to 95–99 years (i.e., 35–39, 40–44, 45–49, 50–54, 55–59, 60–64, 65–69, 70–74, 75–79, 80–84, 85–89, 90–94, and 95–99 years).

### 2.3. Procedures and Statistical Analyses

For statistical analyses, the older age group classes were grouped into one, 90+ years, in order to have enough observations for each event and sex group. In every group, the best eight female and male competitors (i.e., finalists of each age group and event) were recorded. However, eight athletes did not compete in a final in all events and all age groups.

For each competitor the name, age group, nationality, and achieved height or distance was recorded. All data were presented as means ± standard deviations, for continuous variables and as number N (%) for categorical variables. The acceptable type I error was set at *p* < 0.05. All statistical analyses were carried out using a statistical package R, R Core Team (2016). (R: A language and environment for statistical computing, R Foundation for Statistical Computing, Vienna, Austria, URL https://www.R-project.org/).

For data visualization, a ggplot2 package was used. To compare the average performance between genders and within events, a *t*-test was performed. A two-way analysis of variance (ANOVA) was used to compare effects of sex, age groups, and calendar year in performance for each event. Next, effects (sex, time, age group) and interactions (sex and age group) were considered more rigorously through a linear regression model for each event separately. For jumping events, the sex × age interaction was not considered, because the fit of the model was not good. Since repeated measurements occurred within athletes, a mixed model, with random effects on intercept for each athlete, was performed. The R package lmer was used.

The model, for throwing events, was specified as follows:score (Y) ~ (fixed effects (X) = sex + time + age group + sex × age group) + (random effects of intercept = athletes)(1)
and for jumping events:score (Y) ~ (fixed effects (X) = sex + time + age group) + (random effects of intercept = athletes). (2)

Time was modelled in terms of the calendar year being a continuous variable.

In addition, for jumping events, the sex differences in performance were calculated using the formula 100 × (men’s performance–women’s performance)/women’s performance. Instead, a correct comparison of the throwing performances was not possible because male and female master athletes of the same age group used implements of different weight and also female and male athletes of different age groups used implements of different weight [5].

## 3. Results

Overall, 13,673 observations for 4726 master athletes were analyzed. Repeated measurements were observed over years and across events. Overall, 2487 athletes (53% of 4726) participated more than once. There were 1531 athletes (32% of 4726) who participated at different events, on average 2.45 events per athlete over all the periods of observation (the maximum number was seven events per athlete). There were 1988 athletes (42%) who participated several times over the years, on average 3.64 times, the maximum was 15 times. The number of women and men were 1704 and 3022, respectively, with 5786 (42% of 13,673) and 7887 (58%) numbers of observations resulting in an overall men-to-women ratio of 1.36.

For each event, the participation by sex and the men-to-women ratios over time (Figure 1) and by age groups (Figure 2) were reported. It should be noted that, in throwing events, female participation did not change as much over time and the men-to-women ratios were, on average, around one. In the jumping events, the men-to-women ratios were not monotonically decreasing over time with a great variability by event and calendar year. The highest men-to-women ratio values, with an average of 3.3 and a maximum of 6.4, were found in pole vault. For all events, the men-to-women ratios were increasing by age group, with the greatest values for the jumping events. The maximum ratio was of 51, that is 51 men versus one woman in pole vault in the age group 75–79 years.

Table 1 lists the means of performances for each event reported by sex and age groups and by sex and calendar year. For jumping events, the observed sex differences, by calendar year and by age groups, are also shown in Table 1. As expected, for each event, the men always performed on average better than the women performed but the differences were only significant by age groups, and not by years. In shot put only, a sex × calendar year interaction in performance was shown (*p* = 0.024). Overall, the sex differences were as follows: high jump (*p* < 0.001, 15.9%), long jump (*p* < 0.001, 18.6%), pole vault (*p* < 0.001, 31%), and triple jump (*p* < 0.001, 18.1%). A sex × age group interaction in performance was observed (*p* = 0.005 for triple jump and *p* < 0.001 for the rest). The sex differences in performance were very different among the age groups. For all of the events, except triple jump where sex differences were increasing with age, the sex differences were first increasing and then decreasing, reaching a maximum in the age interval 70–79 years (65–69 years in high jump).

Table 2 lists the results of the mixed regression models, described in the methodology and performed for each event separately. For each event, the performances always improved significantly over time (*p* = 0.002 in hammer throw and *p* < 0.001 for the others) except in javelin throw (*p* = 0.369). However, the magnitude of the main effect of calendar year was smaller than the main effect of sex on performance. The male performances were significantly (*p* < 0.001) better than female performances for each event, since the estimates of sex effects, which represented the male-female sex differences, were positive for each event. In jumping events, men performed better than women by ~0.30 m in high jump, ~1.23 m in long jump, ~1.15 m in pole vault, and ~2.58 m in triple jump. The estimates of age effects gave the estimated decline in performances with age. In fact, the first two estimates (i.e., for the age groups 35–39 and 40–44 years) were positive for each event because performances with reference to performances of the age group 45–49 years were compared. Instead, the other coefficients were negative and increasing, meaning that performances declined over age. In throwing events, a different decline of performances with age by sex, given by the estimates of sex × age interaction, was observed. In discus throw, the estimates of this effect were positive and significant, for the age groups between 50 and 84 years, meaning that compared to female performances for these age groups, male performances declined slower over age. Instead, in javelin throw, male performances declined significantly faster in athletes older than 65 years.

It should be noted that for discus throw, javelin throw, and hammer throw, the ages at the highest score were different between the sexes (Table 1 and Table 2) and older ages for men: in discus throw the age groups 60–64 years versus 35–39 years for women, in javelin throw 40–44 years for men and 35–39 years for women, and in hammer throw 50–54 years for men and 40–44 years for women. The lowest score was found in the oldest age groups for all events. Compared to the main effect of calendar year and sex, the main effect of age group had a larger magnitude.

In Figure 3a,b, the decline with age, of the mean score for all athletes and for the world’s record (top score) are shown graphically for each event and by sex. Age was defined as the middle point of each age interval. It was observed for each event that the top female record curve was very close to all male athletes record curve. Top performances for both sexes declined faster with age in jumping events (Figure 3b) and hammer and javelin throw (Figure 3a). However, the decline with age was not monotonic, in particular for throwing events. Figure 4a,b show the difference (%) in performance between males and females by age, as the middle point of each age interval, (Figure 4a) and year (Figure 4b). The lines were not the fitted based on the mixed model, but they were just the smoothing of the plotted values of the differences by event. In triple jump and long jump, the sex differences were increasing with age, even if not monotonically. In pole vault, the sex differences had overall a decreasing, but not monotonic, trend. In high jump, the sex differences did not change as much with age. For all jumping events, sex differences decreased over time.

## 4. Discussion

The main findings of the present study were: (i) men outscored women by 15.9–31% in jumping events; (ii) performance improved linearly by calendar year, but not significantly in javelin throw; (iii) the sex differences in performance changed across calendar years and the trend was different by event; (iv) athletes in younger age groups outscored athletes in older groups; and (v) in triple jump and long jump, the sex differences were increasing with age, even if not monotonically, in pole vault and high jump sex differences had overall a decreasing, but not monotonic, trend.

### 4.1. Men Had a Better Performance than Women in All Events and Age Groups

The quantification of the sex differences in jumping events in master athletes was in agreement with previous studies on other sports relying on muscle power. For instance, maximal muscle power was higher in men than in women wrestlers [23]. Furthermore, hand grip power was greater by ~60–65% in elderly males than in females in various relative loads of maximum voluntary contraction [24]. However, the observed sex differences in the present study were larger than what were previously shown in running events. A study of the World Masters Championships found a sex difference of ~20% [25], which was proportional to the differences between women and men in physiological characteristics such as maximal oxygen uptake [26]. In addition, the variation of sex differences in performance among throwing and jumping events should be attributed to the physiological correlates of each one. Throwing events rely more on muscle power than jumping events which might explain the greater sex differences in performance in the former events.

The decline of male performances with age was slower or faster than the decline of female performances depending on age groups and events. In the World Masters Athletics Championships held from 1975 to 2016, performances declined with age for each event. In addition, for those events, the decline of performances with age was dependent on sex and event [5]. The decline in performances with increasing age seems to be different for throwing and jumping events as compared to running events. An analysis of World Masters Athletics Championships for running events (i.e., 100 m to marathon) found no change in performances from 1975 to 2015 [25].

### 4.2. Younger Athletes Performed Better than Elderly Athletes

The inverse relationship between age group and performance was in agreement with previous studies. Although master throwers have a superior performance than age-matched nonathletes, their muscle strength and power, assessed by squat jump and bench press, declined with aging [27]. Furthermore, it has been suggested that muscle power declines with aging more than muscle strength [28]. Aging had a greater impact on anaerobic than on aerobic power in trained master cyclists [29]. In addition, a decline of muscle mass, and in a lesser degree of the number of motor units, has been observed in master athletes [30]. Previous research had shown that long jump events were mostly affected by aging among both male and female master athletes, whereas the throwing event most affected was the javelin for men and the discus for women [31].

According to these studies, it was not surprising that the lowest score was observed in the oldest age groups for all throwing and jumping events. However, a novel finding was that the age group with the highest score varied by event with jumping events showing the highest scores in the younger age groups as compared with throwing events. It may be assumed that this discrepancy might result from the overall age of peak performance for each event. It has been suggested that the age of peak performance is older in throwers than in jumpers (27.3 versus 25.7 years) [32]. As shown in Table 2, there was an overall linear relationship between performance and age group indicating that the rate of decline did not vary across age.

This observation was in contrast to a previous study reporting that the performance of senior athletes declined by ~3.4% per year over 35 years of competition. Declining slowly from the age of 50 years to the age of 75 years, and dramatically after the age of 75 years [33], which should be attributed to the relatively small number of athletes in the oldest age groups in the present study.

### 4.3. Sex Differences in Performance Depend on the Discipline

It was also found that the trend of sex differences in performance by age were event-dependent and not monotonic (Figure 4a). This finding is in line with recent findings for master swimmers competing in freestyle [10], backstroke [11], breaststroke [12], butterfly [13], individual medley [14], and open-water, long-distance swimming [15] in the FINA World Championships. In these master swimmers, women were able to reduce the gap between women and men with increasing age, where women at the age of ~85 years and older were able to achieve a performance similar to men.

### 4.4. Limitations, Strengths, and Practical Applications

A limitation of the present study was that it considered the master athletes who participated in the European Masters Championships during the last three decades. Their performance may vary from non-European athletes, since performance varies by nationality. Therefore, caution is needed to generalize the findings to non-European athletes. Furthermore, it should be highlighted that the study design was cross-sectional, similar to previous studies [21,22,33]. Therefore, the findings may be affected by selection (survival) bias. Accordingly, the inference for the effect of aging based on age-related differences should be regarded with caution. A further limitation was that the population of male master athletes was greater than the population of female master athletes. Therefore, selecting the top eight athletes from a final means, in fact, to compare the 10% better male athletes with female athletes of “any” quality. This may affect our results. A strength of this study was that it analyzed the top eight finalists for each event in each championship, instead of master world records, resulting in one of the largest data sets ever studied. The large sample size allowed analyzing and drawing safe conclusions about aspects of masters’ performance such as sex, age group, and calendar year. The knowledge of these trends is of importance for athletes and coaches for use in planning a successful career as a master athlete.

## 5. Conclusions

For athletes and coaches, male master athletes achieved a better performance than female master athletes in jumping and throwing events in the European Masters Athletics Championships. Furthermore, the sex differences in performance were increasing with age, even if not monotonically, in triple jump and long jump. However, in pole vault and high jump the sex differences had overall a decreasing, but not monotonic, trend. These findings are of great practical value for coaches and fitness trainers working with master athletes to set realistic goals and to develop optimal training programs. Since these professionals work with athletes of varying sex and age, the knowledge of the sex- and age-related differences of jumping and throwing performances is essential in order to optimize the performance of these master athletes. Furthermore, our findings are of interest for scientists such as geriatrists and sport scientists focusing on the relationship of aging with physical fitness and physical activity. As was mentioned in the introduction, previously, most relevant research either concerned large samples of runners or a limited number of throwers and jumpers. Therefore, our findings filled an important gap in the existing literature and may be used as a reference in future studies. The performance of master athletes changed slowly across years. However, differences in performance by age group were identified (i.e., almost perfect inverse linear relationship between performance and age group) indicating that aging impacts on muscle power to a larger extent than on other components of physical fitness.

## Figures and Tables

**Figure 1 ijerph-16-01200-f001:**
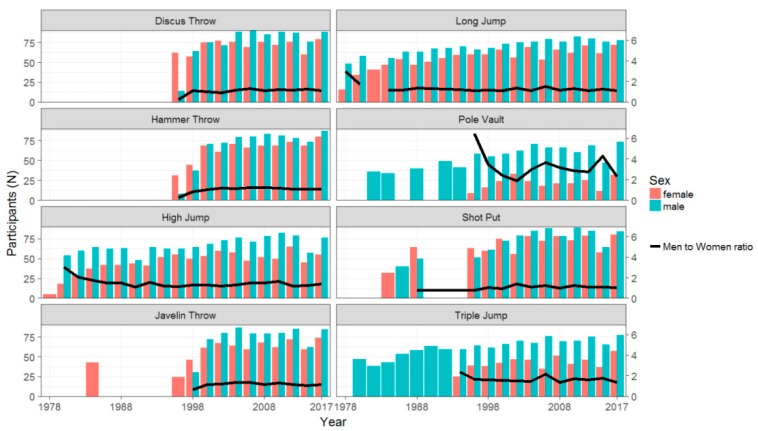
Participation in throwing and jumping events by sex and calendar year.

**Figure 2 ijerph-16-01200-f002:**
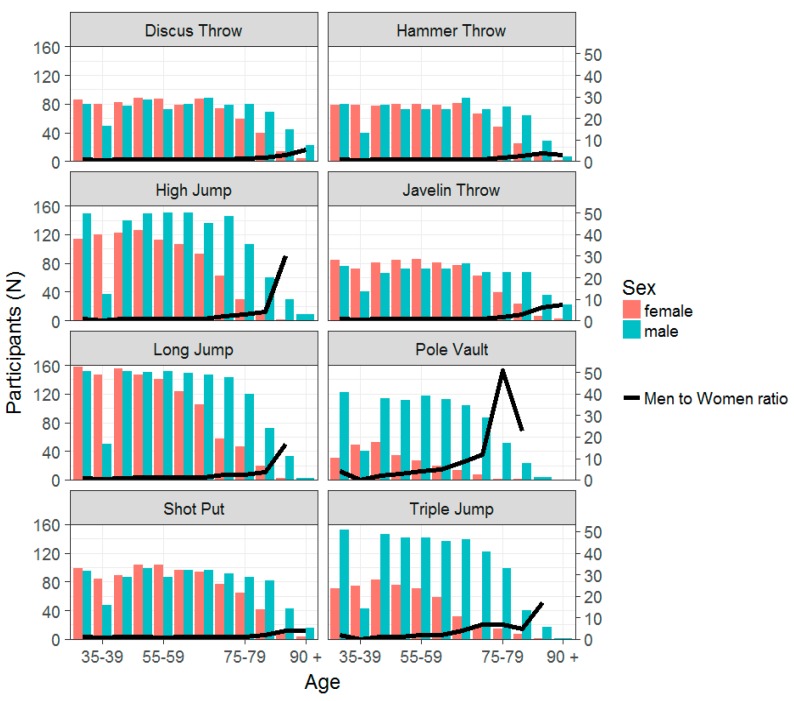
Participation in throwing and jumping events by sex and age-group interval.

**Figure 3 ijerph-16-01200-f003:**
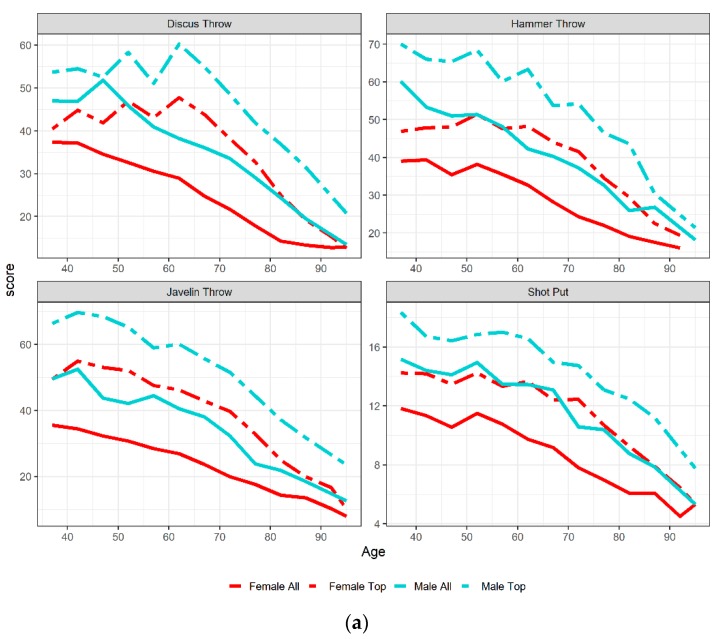
The decline with age of mean score (all athletes) and top score in throwing (**a**) and jumping (**b**) events by sex. Age was defined as a continuous variable, considering the middle of the interval of each age group.

**Figure 4 ijerph-16-01200-f004:**
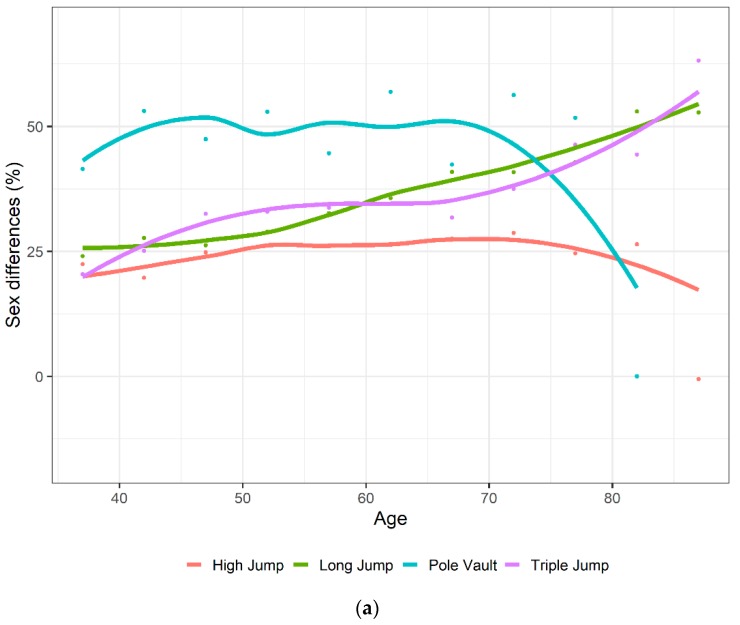
The sex differences (%) in performance of jumping events by age (**a**) and year (**b**). Age was defined as a continuous variable, considering the middle of the interval of each age group. Points are the observed means of the differences of the scores between males and females. Lines were the smoothed values of the observed means.

**Table ijerph-16-01200-t001a:** 

Discus Throw				Hammer Throw				Javelin Throw				Shot Put			
	Score (Mean (SD))				Score (Mean (SD))				Score (Mean (SD))				Score (Mean (SD))		
Overall	Female	Male	*p*	Overall	Female	Male	*p*	Overall	Female	Male	*p*	Overall	Female	Male	*p*
	28.76 (8.56)	38.89 (9.81)	<0.001		32.65 (8.65)	43.01 (10.03)	<0.001		27.85 (8.20)	42.29 (12.59)	<0.001	n	9.85 (2.21)	12.45 (2.32)	<0.001
Age	N (%)	N (%)	<0.001	Age	N (%)	N (%)	<0.001	Age	N (%)	N (%)	<0.001	Age	N (%)	N (%)	<0.001
35–39	37.34 (5.13)	40.43 (6.91)		35–39	39.06 (7.81)	46.88 (11.27)		35–39	35.49 (6.56)	49.43 (11.34)		35–39	11.82 (1.54)	14.24 (1.52)	
40–44	37.14 (5.02)	44.80 (4.26)		40–44	39.38 (5.78)	47.88 (6.91)		40–44	34.41 (6.95)	54.92 (6.98)		40–44	11.35 (1.33)	14.19 (1.11)	
45–49	34.52 (5.60)	41.86 (4.96)		45–49	35.42 (7.22)	48.06 (6.23)		45–49	32.27 (6.04)	53.03 (6.35)		45–49	10.55 (1.32)	13.47 (1.41)	
50–54	32.60 (4.81)	46.92 (4.46)		50–54	38.19 (6.06)	51.42 (7.05)		50–54	30.69 (5.76)	52.09 (4.96)		50–54	11.49 (1.37)	14.24 (1.06)	
55–59	30.57 (4.43)	43.11 (3.10)		55–59	35.54 (5.43)	47.61 (6.29)		55–59	28.44 (4.88)	47.54 (5.90)		55–59	10.76 (1.11)	13.33 (0.88)	
60–64	28.92 (4.43)	47.74 (3.59)		60–64	32.66 (4.57)	48.24 (6.01)		60–64	26.92 (4.97)	46.22 (4.53)		60–64	9.74 (1.49)	13.67 (1.18)	
65–69	24.75 (4.42)	43.80 (3.83)		65–69	28.23 (5.35)	44.05 (4.60)		65–69	23.60 (5.18)	42.87 (4.19)		65–69	9.17 (1.39)	12.40 (1.04)	
70–74	21.66 (3.84)	38.13 (3.52)		70–74	24.32 (4.76)	41.56 (5.50)		70–74	19.95 (4.71)	39.73 (4.82)		70–74	7.81 (1.20)	12.45 (0.95)	
75–79	17.85 (3.34)	32.76 (3.78)		75–79	22.02 (4.65)	34.58 (4.72)		75–79	17.61 (3.48)	32.89 (6.70)		75–79	6.98 (1.38)	10.69 (1.38)	
80–84	14.28 (3.57)	25.05 (4.67)		80–84	19.13 (3.85)	29.50 (6.39)		80–84	14.34 (4.12)	24.98 (5.04)		80–84	6.08 (1.12)	9.26 (1.50)	
85–89	13.31 (2.80)	19.20 (4.04)		85–89	17.57 (5.35)	22.53 (4.53)		85–89	13.53 (3.48)	19.85 (5.01)		85–89	6.08 (1.21)	7.90 (1.36)	
90 +	12.79 (0.67)	14.95 (2.83)		90 +	16.02 (3.10)	19.39 (1.78)		90 +	9.50 (2.69)	16.07 (3.86)		90 +	4.71 (0.77)	6.24 (1.01)	
Year	N (%)	N (%)	0.849	Year	N (%)	N (%)	0.522	Year	N (%)	N (%)	0.367	Year	N (%)	N (%)	0.024
-	-	-		-	-	-		1984	27.42 (7.99)	-		1984	8.78 (1.42)	-	
-	-	-		-	-	-		-	-	-		1986	-	13.47 (1.13)	
-	-	-		-	-	-		-	-	-		1988	9.47 (2.20)	11.71 (2.41)	
1996	29.60 (9.23)	43.71 (5.65)		1996	28.95 (7.34)	42.49 (7.58)		1996	23.44 (9.25)	-		1996	10.03 (2.19)	12.51 (1.92)	
1998	31.45 (7.35)	39.84 (9.52)		1998	35.54 (6.66)	42.94 (10.09)		1998	30.09 (7.90)	38.29 (13.61)		1998	10.50 (2.11)	12.76 (2.19)	
2000	27.62 (8.50)	39.76 (8.88)		2000	30.27 (8.49)	43.65 (10.09)		2000	26.63 (7.93)	42.54 (13.16)		2000	9.58 (2.31)	12.64 (2.00)	
2002	29.39 (9.66)	39.98 (9.76)		2002	34.11 (7.93)	43.88 (10.31)		2002	29.73 (8.21)	42.52 (13.94)		2002	11.15 (1.29)	12.09 (2.62)	
2004	28.31 (8.45)	38.91 (10.12)		2004	31.41 (8.86)	43.58 (10.28)		2004	28.31 (8.47)	42.47 (13.10)		2004	9.68 (2.31)	12.57 (2.55)	
2006	28.08 (8.04)	37.84 (9.95)		2006	32.64 (9.46)	42.12 (10.08)		2006	29.45 (8.47)	42.82 (12.79)		2006	9.70 (2.14)	12.36 (2.33)	
2008	28.88 (8.67)	38.90 (10.07)		2008	32.93 (8.56)	42.28 (9.93)		2008	27.96 (7.43)	42.51 (11.54)		2008	9.82 (2.52)	12.75 (2.53)	
2010	28.58 (8.08)	38.36 (10.88)		2010	32.15 (8.31)	41.94 (10.58)		2010	26.57 (9.11)	42.12 (11.96)		2010	9.65 (2.07)	12.14 (2.41)	
2012	28.82 (8.34)	38.53 (10.13)		2012	35.26 (8.95)	45.15 (9.19)		2012	27.65 (7.47)	43.85 (12.39)		2012	10.11 (2.35)	12.58 (2.37)	
2014	28.12 (8.55)	37.66 (8.97)		2014	31.88 (7.87)	41.95 (9.55)		2014	25.87 (6.79)	41.00 (11.20)		2014	9.39 (2.36)	11.96 (2.31)	
2017	28.13 (8.83)	38.83 (9.93)		2017	33.02 (9.58)	42.79 (10.45)		2017	28.53 (8.76)	41.97 (12.65)		2017	9.87 (2.03)	12.56 (2.15)	

**Table ijerph-16-01200-t001b:** 

Long Jump					High Jump					Pole Vault					Triple Jump				
	Score (Mean (SD))					Score (Mean (SD))					Score (Mean (SD))					Score (Mean (SD))			
Overall	Female	Male	diff%	*p*	Overall	Female	Male	diff%	*p*	Overall	Female	Male	diff%	*p*	Overall	Female	Male	diff%	*p*
	4.26 (0.87)	5.06 (1.07)	18.57	<0.001		1.30 (0.20)	1.50 (0.24)	15.94	<0.001		2.49 (0.50)	3.23 (0.70)	31.01	<0.001		9.11 (1.70)	10.76 (2.15)	18.13	<0.001
Age	N (%)	N (%)	diff%	<0.001	Age	N (%)	N (%)	diff%	<0.001	Age	N (%)	N (%)	diff%	<0.001	Age	N (%)	N (%)	diff%	0.005
35–39	5.23 (0.38)	6.41 (0.37)	22.56		35–39	1.53 (0.09)	1.85 (0.11)	20.64		35–39	2.79 (0.46)	3.97 (0.46)	42.29		35–39	10.95 (0.95)	13.48 (0.95)	23.18	
40–44	5.01 (0.32)	6.37 (0.28)	27.14		40–44	1.50 (0.10)	1.79 (0.10)	19.22		40–44	2.67 (0.48)	3.95 (0.47)	48.04		40–44	10.46 (0.83)	13.10 (0.84)	25.18	
45–49	4.75 (0.40)	5.98 (0.37)	26.02		45–49	1.39 (0.11)	1.73 (0.09)	24.26		45–49	2.62 (0.38)	3.82 (0.38)	46.03		45–49	9.69 (0.95)	12.64 (0.85)	30.4	
50–54	4.42 (0.36)	5.74 (0.34)	29.9		50–54	1.32 (0.10)	1.65 (0.10)	25.54		50–54	2.44 (0.43)	3.54 (0.37)	45.09		50–54	9.03 (0.95)	11.86 (0.87)	31.27	
55–59	4.11 (0.35)	5.39 (0.30)	31.07		55–59	1.26 (0.10)	1.57 (0.10)	24.77		55–59	2.38 (0.31)	3.29 (0.42)	38.36		55–59	8.63 (0.88)	11.34 (0.89)	31.38	
60–64	3.78 (0.47)	5.01 (0.38)	32.83		60–64	1.19 (0.10)	1.49 (0.09)	25.03		60–64	2.12 (0.33)	3.07 (0.33)	44.98		60–64	7.96 (0.79)	10.53 (0.81)	32.36	
65–69	3.39 (0.46)	4.66 (0.35)	37.57		65–69	1.11 (0.09)	1.41 (0.09)	27.12		65–69	1.96 (0.35)	2.75 (0.29)	40.14		65–69	7.47 (0.74)	9.60 (0.94)	28.43	
70–74	3.11 (0.45)	4.22 (0.40)	35.8		70–74	1.03 (0.08)	1.31 (0.09)	26.82		70–74	1.69 (0.27)	2.50 (0.29)	48.24		70–74	6.46 (0.83)	8.72 (0.96)	34.86	
75–79	2.65 (0.40)	3.74 (0.42)	41.02		75–79	0.99 (0.08)	1.21 (0.08)	21.98		75–79	1.45	2.21 (0.26)	52.39		75–79	5.64 (0.92)	7.77 (0.98)	37.68	
80–84	2.33 (0.42)	3.24 (0.48)	39.05		80–84	0.89 (0.07)	1.11 (0.09)	24.12		80–84	1.40	1.86 (0.26)	33.07		80–84	4.57 (0.63)	6.83 (1.04)	49.55	
85–89	2.16 (0.78)	2.83 (0.54)	30.8		85–89	0.90	1.01 (0.10)	12.41		85–89	-	1.75 (0.26)	-		85–89	3.16	5.95 (0.99)	88.4	
90 +	-	2.35 (0.87)	-		90 +	-	0.95 (0.07)	-		90 +	-	-	-		90 +	-	3.31	-	
Year	N (%)	N (%)	diff%	0.855	Year	N (%)	N (%)	diff%	0.386	Year	N (%)	N (%)	diff%	0.306	Year	N (%)	N (%)	diff%	0.828
1978	4.71 (0.55)	5.14 (0.75)	9.14		1978	1.32 (0.19)	-	-		-	-	-	-		-	-	-	-	
1980	4.20 (0.71)	4.96 (1.07)	17.91		1980	1.25 (0.16)	1.43 (0.20)	13.71		-	-	-	-		1980	-	10.39 (1.92)	-	
1982	4.52 (0.90)	-	-		1982	1.34 (0.17)	1.48 (0.22)	10.05		1982	-	3.29 (0.55)	-		1982	-	11.27 (1.62)	-	
1984	4.22 (0.84)	5.13 (0.96)	21.55		1984	1.27 (0.18)	1.44 (0.21)	13.75		1984	-	2.99 (0.57)	-		1984	-	10.99 (1.99)	-	
1986	4.23 (0.85)	5.20 (0.92)	23.05		1986	1.29 (0.20)	1.49 (0.23)	15.38		-	-	-	-		1986	-	10.86 (2.11)	-	
1988	4.41 (0.82)	5.09 (0.98)	15.43		1988	1.29 (0.20)	1.50 (0.23)	16.4		1988	-	3.25 (0.65)	-		1988	-	10.60 (2.02)	-	
1990	4.42 (0.89)	5.17 (1.05)	16.88		1990	1.30 (0.23)	1.62 (0.22)	24.33		-	-	-	-		1990	-	10.77 (2.03)	-	
1992	4.18 (0.86)	5.11 (1.02)	22.37		1992	1.32 (0.20)	1.53 (0.21)	15.95		1992	-	3.25 (0.61)	-		1992	-	10.63 (1.95)	-	
1994	4.27 (0.81)	5.05 (1.02)	18.3		1994	1.27 (0.21)	1.54 (0.22)	20.67		1994	-	3.53 (0.64)	-		1994	9.41 (1.34)	10.90 (1.97)	15.78	
1996	4.32 (0.91)	5.16 (0.98)	19.27		1996	1.29 (0.21)	1.55 (0.22)	19.58		1996	2.16 (0.42)	3.28 (0.78)	52.21		1996	9.13 (1.56)	10.98 (2.12)	20.36	
1998	4.29 (0.81)	5.05 (1.02)	17.62		1998	1.33 (0.17)	1.50 (0.20)	12.96		1998	2.25 (0.37)	3.21 (0.72)	42.6		1998	8.83 (1.58)	11.08 (2.03)	25.42	
2000	4.11 (0.90)	4.99 (1.11)	21.26		2000	1.31 (0.20)	1.49 (0.23)	14.08		2000	2.33 (0.34)	3.27 (0.70)	40.36		2000	8.98 (1.67)	10.50 (2.02)	16.89	
2002	4.46 (0.75)	4.88 (1.09)	9.26		2002	1.31 (0.20)	1.50 (0.26)	14.8		2002	2.49 (0.37)	3.25 (0.79)	30.51		2002	9.19 (1.71)	10.46 (2.46)	13.84	
2004	4.09 (1.00)	4.93 (1.16)	20.62		2004	1.31 (0.19)	1.47 (0.27)	12.12		2004	2.57 (0.56)	3.25 (0.77)	26.61		2004	9.00 (1.94)	10.77 (2.31)	19.58	
2006	4.31 (0.94)	5.17 (1.12)	19.98		2006	1.36 (0.17)	1.50 (0.25)	10.1		2006	2.50 (0.51)	3.12 (0.76)	24.62		2006	8.75 (1.89)	10.67 (2.49)	21.93	
2008	4.23 (0.86)	5.15 (1.12)	21.71		2008	1.28 (0.22)	1.51 (0.27)	17.59		2008	2.38 (0.66)	3.26 (0.68)	37.16		2008	8.89 (1.73)	10.86 (2.24)	22.16	
2010	4.32 (0.90)	4.95 (1.17)	14.66		2010	1.30 (0.21)	1.51 (0.28)	16.16		2010	2.59 (0.63)	3.26 (0.64)	26.06		2010	9.34 (1.73)	10.60 (2.29)	13.46	
2012	4.20 (0.89)	5.07 (1.16)	20.79		2012	1.29 (0.21)	1.50 (0.28)	16.21		2012	2.63 (0.51)	3.29 (0.66)	25.05		2012	9.32 (1.72)	10.96 (2.14)	17.57	
2014	4.05 (0.84)	4.96 (1.23)	22.3		2014	1.25 (0.22)	1.49 (0.26)	18.71		2014	2.60 (0.40)	2.93 (0.62)	12.6		2014	9.34 (1.60)	10.58 (2.20)	13.27	
2017	4.24 (0.87)	5.12 (1.11)	20.73		2017	1.29 (0.19)	1.55 (0.25)	20.22		2017	2.65 (0.48)	3.27 (0.72)	23.32		2017	9.19 (1.77)	10.79 (2.39)	17.31	

**Table 2 ijerph-16-01200-t002:** The estimates and *p*-values of fixed effects of mixed models of performance for each event. The effects on sex are reported with women being the reference sex group and the effects on age are shown with 45–49 years being the reference age group.

	Discus Throw	Hammer Throw	Javelin Throw	Shot Put	High Jump	Long Jump	Triple Jump	Pole Vault
	N = 1605, Athletes = 811	N = 1451, Athletes = 720	N = 1436, Athletes = 774	N = 1794, Athletes = 953	N = 2163, Athletes = 1079	N = 2421, Athletes = 1266	N = 1684, Athletes = 832	N = 1119, Athletes = 519
	Estimate	Pr (>|t|)	Estimate	Pr (>|t|)	Estimate	Pr (>|t|)	Estimate	Pr (>|t|)	Estimate	Pr (>|t|)	Estimate	Pr (>|t|)	Estimate	Pr (>|t|)	Estimate	Pr (>|t|)
(Intercept)	−66.062	0.032	−102.969	0.019	−5.063	0.897	−42.778	<0.001	−5.865	<0.001	−9.091	<0.001	−21.428	<0.001	−14.666	<0.001
Year	0.050	0.001	0.068	0.002	0.018	0.369	0.026	<0.001	0.004	<0.001	0.007	<0.001	0.016	<0.001	0.009	<0.001
Sex = male	7.644	<0.001	11.715	<0.001	20.707	<0.001	2.942	<0.001	0.305	<0.001	1.235	<0.001	2.580	<0.001	1.147	<0.001
Age 35–39	3.283	<0.001	3.199	<0.001	4.088	<0.001	1.213	<0.001	0.122	<0.001	0.486	<0.001	0.978	<0.001	0.199	<0.001
Age 40–44	1.933	<0.001	2.480	<0.001	2.428	<0.001	0.655	<0.001	0.075	<0.001	0.300	<0.001	0.568	<0.001	0.133	<0.001
Age 50–54	−1.721	<0.001	2.286	<0.001	−1.204	0.025	0.752	<0.001	−0.082	<0.001	−0.313	<0.001	−0.744	<0.001	−0.273	<0.001
Age 55–59	−4.208	<0.001	−0.291	0.641	−2.991	<0.001	−0.077	0.511	−0.154	<0.001	−0.638	<0.001	−1.292	<0.001	−0.478	<0.001
Age 60–64	−6.755	<0.001	−3.366	<0.001	−4.651	<0.001	−0.911	<0.001	−0.226	<0.001	−0.971	<0.001	−2.042	<0.001	−0.688	<0.001
Age 65–69	−10.101	<0.001	−6.686	<0.001	−6.749	<0.001	−1.518	<0.001	−0.304	<0.001	−1.329	<0.001	−2.755	<0.001	−0.916	<0.001
Age 70–74	−13.319	<0.001	−10.145	<0.001	−9.868	<0.001	−2.587	<0.001	−0.393	<0.001	−1.740	<0.001	−3.609	<0.001	−1.192	<0.001
Age 75–79	−15.727	<0.001	−10.606	<0.001	−11.771	<0.001	−2.982	<0.001	−0.481	<0.001	−2.155	<0.001	−4.457	<0.001	−1.469	<0.001
Age 80–84	−17.849	<0.001	−12.352	<0.001	−13.619	<0.001	−3.801	<0.001	−0.594	<0.001	−2.645	<0.001	−5.501	<0.001	−1.773	<0.001
Age 85–89	−19.150	<0.001	−11.333	<0.001	−15.868	<0.001	−4.092	<0.001	−0.700	<0.001	−3.111	<0.001	−6.237	<0.001	−1.900	<0.001
Age 90 +	−20.648	<0.001	−12.691	<0.001	−17.709	<0.001	−4.948	<0.001	−0.807	<0.001	−3.526	<0.001	−8.044	<0.001		<0.001
Sex = male: age 35–39	−2.977	<0.001	−1.061	0.333	−4.416	<0.001	−0.323	0.157								
Sex = male: age 40–44	−0.372	0.545	−1.272	0.116	−0.603	0.482	0.060	0.730								
Sex = male: age 50–54	5.829	<0.001	1.334	0.085	0.455	0.568	−0.122	0.452								
Sex = male: age 55–59	4.794	<0.001	−0.290	0.738	−1.204	0.165	−0.113	0.522								
Sex = male: age 60–64	12.666	<0.001	3.064	0.001	−1.742	0.050	0.963	<0.001								
Sex = male: age 65–69	11.720	<0.001	2.366	0.014	−4.134	<0.001	0.327	0.085								
Sex = male: age 70–74	9.272	<0.001	3.844	<0.001	−4.005	<0.001	1.420	<0.001								
Sex = male: age 75–79	5.927	<0.001	−2.042	0.065	−7.696	<0.001	0.120	0.573								
Sex = male: age 80–84	1.776	0.042	−3.190	0.011	−11.233	<0.001	−0.045	0.847								
Sex = male: age 85–89	−1.309	0.220	−10.045	<0.001	−13.653	<0.001	−1.100	0.001								
Sex = male: age 90 +	−4.362	0.013	−10.937	<0.001	−17.046	<0.001	−1.564	0.005

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
