# Peer review of "Changes in Jumping and Throwing Performances in Age-Group Athletes Competing in the European Masters Athletics Championships between 1978 and 2017"

_ijerph, 2019, doi:10.3390/ijerph16071200_

Round 1

Reviewer 1 Report

Overall interesting paper about a topic that is not well studied.  There are a few concerns with the figures (see comments below) and one of the tables.

Table 1: please check the numbers---some are in European format (comma instead of a period); also, there is a notation (%) after the headings of 'age' and 'year' yet the values in that part of the table do not appear to be %'s.  Please clarify.

Table 2 is extremely confusing and really does not add anything to the manuscript.  Please consider removing this table and any reference to it within the paper.

Figures 3.1-3.8 and 4.1-4.8 are easy to understand but do not add anything to the paper that hasn't already been presented in some other table or figure.  Also, figure 4.a is not really necessary either and is not as important as 4.b to the paper.  Please consider removing Figures 3.1-3.8 and 4.1-4.8, and 4.a and any reference to them within the paper.

Author Response

Open Review

English language and style

( ) Extensive editing of English language and style required
( ) Moderate English changes required
(x) English language and style are fine/minor spell check required
( ) I don't feel qualified to judge about the English language and style

Yes

Can be improved

Must be improved

Not applicable

Does the   introduction provide sufficient background and include all relevant   references?

(x)

( )

( )

( )

Is the   research design appropriate?

(x)

( )

( )

( )

Are the   methods adequately described?

(x)

( )

( )

( )

Are the   results clearly presented?

( )

( )

(x)

( )

Are the   conclusions supported by the results?

(x)

( )

( )

( )

Comments and Suggestions for Authors

Overall interesting paper about a topic that is not well studied.  There are a few concerns with the figures (see comments below) and one of the tables.

Table 1: please check the numbers---some are in European format (comma instead of a period); also, there is a notation (%) after the headings of 'age' and 'year' yet the values in that part of the table do not appear to be percent.  Please clarify.

Answer: We thank the reviewer for having noted that. We have modified the table accordingly.

Table 2 is extremely confusing and really does not add anything to the manuscript.  Please consider removing this table and any reference to it within the paper.

Answer: We thank the reviewer for this comment. However, we think that this table is very important because it shows the results of the regression models. Maybe the reviewer claimed this because the results reported in this table were not clearly highlighted in the results section and so we have done that.

Figures 3.1-3.8 and 4.1-4.8 are easy to understand but do not add anything to the paper that hasn't already been presented in some other table or figure.  Also, figure 4.a is not really necessary either and is not as important as 4.b to the paper.  Please consider removing Figures 3.1-3.8 and 4.1-4.8, and 4.a and any reference to them within the paper.

Answer: We agree with the expert reviewer. We removed Figures 3.1-3.8 and 4.1-4.8 and changed the others.

Reviewer 2 Report

This is an interesting manuscript characterise the age and sex related difference in muscle power in master athletes. The authors indicate that aging impacts on muscle power in a larger extent than on other components of physical fitness. Furthermore the authors report that the sex difference in performance increased with increasing age but decreased then after the age of ~70 years. The methodology is appriopriate. However, the authors do not give enough background what already is know in the scientific literature  about age/sex - related physiological determinants of muscle power. The graphic presentation of the results is very extensive, but do not give information about age-related performance kinetic. What is the percent of performance decrease is important to show, especially that the authors compare their own results to the other papers (Pg. 20, paragraph 287-290). What is the reason of different pattern of difference between male and female by age in the Triple Jump after 70 years of age - the authors should be explained.

Additional comments:

1) Pg. 4, there is additional character "n" without data coverage in table 1. kolumn 16; line 4.

Suggestion: delete this "n" sign.

2) Pg. 8, figure 1. "Participation in throwing and jumping events by sex and calendar year" and figure 2. "Participation in throwing and jumping events by sex and age-group" - are unecessalily used.

Explanation: There is suitable data description about participation by sex and men-to-women ratios over time and by age groups in paragraph 142-149 (Pg.7). Additionally the aim of the present study wasn't to investigate participation but the performance trends by age, sex and calendar year in master athletes - as the authors describe in paragraph 73-76 (Pg.2).

3) Pg. 10-18, figure 3. (3.1-3.8) (Pg. "Performance in throwing and jumping events by sex, calendar year and age group" and figure 4. (4.1-4.8) "Sex difference in performance in throwing and jumping events by sex, calendar year and age group" -  are unnecessalily used.

Explanation: All the effects for each events separately (using mixed model) are described in Table 2. In this case the graphical visualization of data seems unnecessary duplicated. In this place it would be interesting to show the cut-off points for performance as a results of each events separately - of course in graphical form.

Author Response

Open Review

English language and style

( ) Extensive editing of English language and style required
( ) Moderate English changes required
( ) English language and style are fine/minor spell check required
(x) I don't feel qualified to judge about the English language and style

Yes

Can be improved

Must be improved

Not applicable

Does the   introduction provide sufficient background and include all relevant   references?

(x)

( )

( )

( )

Is the   research design appropriate?

(x)

( )

( )

( )

Are the   methods adequately described?

(x)

( )

( )

( )

Are the   results clearly presented?

( )

(x)

( )

( )

Are the   conclusions supported by the results?

(x)

( )

( )

( )

Comments and Suggestions for Authors

This is an interesting manuscript characterize the age and sex related difference in muscle power in master athletes. The authors indicate that aging impacts on muscle power in a larger extent than on other components of physical fitness. Furthermore the authors report that the sex difference in performance increased with increasing age but decreased then after the age of ~70 years. The methodology is appropriate. However, the authors do not give enough background what already is known in the scientific literature about age/sex - related physiological determinants of muscle power. The graphic presentation of the results is very extensive, but do not give information about age-related performance kinetic. What is the percent of performance decrease is important to show, especially that the authors compare their own results to the other papers (Pg. 20, paragraph 287-290). What is the reason of different pattern of difference between male and female by age in the Triple Jump after 70 years of age - the authors should be explained.

Answer: We agree with the expert reviewer and improved the discussion to be more precise for jumping and throwing in master athletes. Furthermore, the figures are now reduced.

Additional comments:

1) Pg. 4, there is additional character "n" without data coverage in table 1. kolumn 16; line 4.

Suggestion: delete this "n" sign.

Answer: We thank the reviewer. We have corrected the typo.

2) Pg. 8, figure 1. "Participation in throwing and jumping events by sex and calendar year" and figure 2. "Participation in throwing and jumping events by sex and age-group" - are unnecessarily used.

Answer: We adapted the legend accordingly.

Explanation: There is suitable data description about participation by sex and men-to-women ratios over time and by age groups in paragraph 142-149 (Pg.7). Additionally the aim of the present study wasn't to investigate participation but the performance trends by age, sex and calendar year in master athletes - as the authors describe in paragraph 73-76 (Pg.2).

Answer: We thank the reviewer for this comment. However, we think that a description of participation and a graphical visualization is necessary. (Beat check if right)

3) Pg. 10-18, figure 3. (3.1-3.8) (Pg. "Performance in throwing and jumping events by sex, calendar year and age group" and figure 4. (4.1-4.8) "Sex difference in performance in throwing and jumping events by sex, calendar year and age group" -  are unnecessarily used.

Answer: We adapted the legend accordingly.

Explanation: All the effects for each events separately (using mixed model) are described in Table 2. In this case the graphical visualization of data seems unnecessary duplicated. In this place it would be interesting to show the cut-off points for performance as a result of each event separately - of course in graphical form.

Answer: We thank the reviewer for this comment. We have removed these figures and added the figures showing the decline of performance with age for each event.

Reviewer 3 Report

I understand the need for such research, however, I feel quite a few major things need to be addressed before acceptance. Apologies for coming across as critical, however, it tends to happen in these type of reviews. I'll keep the comments relatively general and provide more specific feedback if provided the oppurtunity to address the following changes. 

The use of the term "power"

Throughout the manuscript, the term muscle power in my opinion is used incorrectly. Power is a mechanical term that is the rate at which work is produced (work/time). Within this study, the term muscular power is used to describe the performance of specific field events in athletics. These distances thrown or jumped are not examples of muscular power, but performance measures that may be influenced by power. Due to this, links made in the introduction of low muscular power making daily activities more difficult are not addressed in the methodology, as there is no test of actual power. 

Title

I feel the title needs to change. Age-related differences, however, most of the discussion focusses on gender. The term power is used, although no measure of power is actually completed. Jumping and throwing performance, the study uses track events for athletics. Maybe the title can be that specific.

Figures and tables 

A lot of the figures are just repeating information from previous figures or tables and don't add anything to the results section. For example, both males and females a graphically represented for each field event (total score). Then, in seperate figures, the percentage difference between males and females is shown graphically, however, we can already see the difference (total not %) in earlier figures. 

Intent of the study

I feel you are trying to answer too much in the one study. From the title, my underdstanding was that you were going to get averages for each age group across all events. I don't understand why all of the figures in 3 are presented? Why do we want to compare scores from 1996 to 2017? It seems that all of this figures do not answer the research questions and i'm not sure why they are there. Yes, people have improved from 1996 to 2017, but this is the majority of the results section and I don't think it adds all that much. Should there not be figures that compare the different age groups for a particular event? 

Similar to the above point, men outscored women. This doesn't exactly provide that much information, yet it makes up a lot of the results and discussion sections. 

Writing

For this kind of research, I feel thye writing should not be written in 1st person. It is my opinion that the authors should avoid the terms "we" or "our" throughout the manucript and rather write in 3rd person.

Good luck

Author Response

Open Review

English language and style

( ) Extensive editing of English language and style required
(x) Moderate English changes required
( ) English language and style are fine/minor spell check required
( ) I don't feel qualified to judge about the English language and style

Yes

Can be improved

Must be improved

Not applicable

Does the   introduction provide sufficient background and include all relevant   references?

( )

( )

(x)

( )

Is the   research design appropriate?

( )

( )

(x)

( )

Are the   methods adequately described?

( )

(x)

( )

( )

Are the   results clearly presented?

( )

( )

(x)

( )

Are the   conclusions supported by the results?

( )

( )

(x)

( )

Comments and Suggestions for Authors

I understand the need for such research; however, I feel quite a few major things need to be addressed before acceptance. Apologies for coming across as critical, however, it tends to happen in these types of reviews. I'll keep the comments relatively general and provide more specific feedback if provided the opportunity to address the following changes. 

The use of the term "power"

Throughout the manuscript, the term muscle power in my opinion is used incorrectly. Power is a mechanical term that is the rate at which work is produced (work/time). Within this study, the term muscular power is used to describe the performance of specific field events in athletics. These distances thrown or jumped are not examples of muscular power, but performance measures that may be influenced by power. Due to this, links made in the introduction of low muscular power making daily activities more difficult are not addressed in the methodology, as there is no test of actual power. 

Answer: We agree with the expert reviewer and changed in the text to indicate that we investigate and compare throwing and jumping performances in master athletes.

Title

I feel the title needs to change. Age-related differences, however, most of the discussion focusses on gender. The term power is used, although no measure of power is actually completed. Jumping and throwing performance, the study uses track events for athletics. Maybe the title can be that specific.

Answer: We agree with the expert reviewer and changed the title to ‘Changes in jumping and throwing performance in age-group athletes competing in the European Championships 1978-2017’

Figures and tables 

A lot of the figures are just repeating information from previous figures or tables and don't add anything to the results section. For example, both males and females are graphically represented for each field event (total score). Then, in separate figures, the percentage difference between males and females is shown graphically; however, we can already see the difference (total not %) in earlier figures. 

Answer: We agree with the reviewer and we have removed some figures.

Intent of the study

I feel you are trying to answer too much in the one study. From the title, my understanding was that you were going to get averages for each age group across all events. I don't understand why all of the figures in 3 are presented? Why do we want to compare scores from 1996 to 2017? It seems that all of these figures do not answer the research questions and I’m not sure why they are there. Yes, people have improved from 1996 to 2017, but this is the majority of the results section and I don't think it adds all that much. Should there not be figures that compare the different age groups for a particular event? 

Similar to the above point, men outscored women. This doesn't exactly provide that much information, yet it makes up a lot of the results and discussion sections. 

Answer: We agree with the reviewer and we have added the figures showing the decline of performance with age for jumping and throwing events.

Writing

For this kind of research, I feel the writing should not be written in 1st person. It is my opinion that the authors should avoid the terms "we" or "our" throughout the manuscript and rather write in 3rd person.

Answer: We agree with the reviewer and we have changed the manuscript accordingly.

Good luck

Answer: Thank you

Round 2

Reviewer 3 Report

Thank you for taking on board my original comments. All authors should be commmended on how you have addressed these. I've come up with a few more minor comments below.

Line 67: Can you provide an example of ther previous sample sizes used?  That way, it can be compared later in the manuscript to the increased sample size that has been used in this present investigation

Line 284: The comment "almost perfect linear relationship" is probably a little strong. Potentially don't be so bold with this statement.

Author Response

Comments and Suggestions for Authors

Thank you for taking on board my original comments. All authors should be commmended on how you have addressed these. I've come up with a few more minor comments below.

Line 67: Can you provide an example of the previous sample sizes used?  That way, it can be compared later in the manuscript to the increased sample size that has been used in this present investigation

Answer: We agree with the expert reviewer and changed to ‘A research gap in the existing literature was the lack of research in master athletes of jumping and throwing sports disciplines using larger samples than in the past [17,18]. For example, Gava et al. [17] investigated in 2015 the World records in running events by the end of 2012 in 100m, 200m, 400m, 800m, 1500m, 3000m, 5000m, 10000m in athletes from 35 to 100 years. And Baker et al. [18] analysed the Masters athletic records in track and field events, published in September 1999, for mature age athletes over the age of 30 years by decade. The track events were divided in the spring events (i.e. 100m, 200m, and 400m), middle distance events (i.e. 800m, 1500m, and 3000m), long distance (i.e. 5000m, 10000m, and marathon), hurdling events (i.e. 110m, 400m, and steeple chase) and walking events. The jumping events (i.e. long jump, high jump, triple jump, and pole vault) were separated from the throwing events (i.e. shot put, discus, javelin, and hammer).

Line 284: The comment "almost perfect linear relationship" is probably a little strong. Potentially don't be so bold with this statement.

Answer: We agree with the expert reviewer and changed to ‘As shown in Table 2, there was an overall linear relationship between performance and age group indicating that the rate of decline did not vary across age’